# Infrared nanoscopy and tomography of intracellular structures

Katerina Kanevche [1], David J. Burr[2], Dennis J. Nürnberg [3], Pascal K. Hass[4], Andreas Elsaesser [2✉] & Joachim Heberle [1✉]

Although techniques such as fluorescence-based super-resolution imaging or confocal microscopy simultaneously gather both morphological and chemical data, these techniques often rely on the use of localized and chemically specific markers. To eliminate this flaw, we have developed a method of examining cellular cross sections using the imaging power of scattering-type scanning near-field optical microscopy and Fourier-transform infrared spectroscopy at a spatial resolution far beyond the diffraction limit. Herewith, nanoscale surface and volumetric chemical imaging is performed using the intrinsic contrast generated by the characteristic absorption of mid-infrared radiation by the covalent bonds. We employ infrared nanoscopy to study the subcellular structures of eukaryotic (*Chlamydomonas reinhardtii*) and prokaryotic (*Escherichia coli*) species, revealing chemically distinct regions within each cell such as the microtubular structure of the flagellum. Serial 100 nm-thick cellular cross-sections were compiled into a tomogram yielding a three-dimensional infrared image of subcellular structure distribution at 20 nm resolution. The presented methodology is able to image biological samples complementing current fluorescence nanoscopy but at less interference due to the low energy of infrared radiation and the absence of labeling.

[1] Freie Universität Berlin, Department of Physics, Experimental Molecular Biophysics, Arnimallee 14, 14195 Berlin, Germany. [2] Freie Universität Berlin, Department of Physics, Experimental Biophysics and Space Sciences, Arnimallee 14, 14195 Berlin, Germany. [3] Freie Universität Berlin, Department of Physics, Biochemistry and Biophysics of Photosynthetic Organisms, Arnimallee 14, 14195 Berlin, Germany. [4] Freie Universität Berlin, Department of Veterinary Medicine, Institute of Veterinary Anatomy Koserstr. 20, 14195 Berlin, Germany. ✉email: a.elsaesser@fu-berlin.de; joachim.heberle@fu-berlin.de

Cutting edge single-cell techniques can allow for highly precise detection of subcellular components. Observation of the molecular composition of such organelles provides further understanding of their underlying biological processes, cellular mechanics and structural interactions. The visualization of subcellular components is commonly achieved by fluorescence or electron microscopy (EM). Despite the resolving power of cryoEM[1], this method provides little to no information about the chemical composition of the sample. In contrast, colorimetric or fluorescent markers chemically interact with specific cellular components, and thus can produce particularly detailed images of complex intracellular structures[2]. Although fluorescence-based super-resolution microscopy offers the possibility to perform biological and biomedical imaging beyond the diffraction limit and has potential to be furthered by novel technological developments[3], fluorescence imaging remains reliant on the use of chemically-specific labels.

Vibrational spectroscopy and microscopy overcome this barrier, performing chemical imaging in a label-free manner. The sensitivity and speed of infrared (IR) microscopic imaging of biological samples has been dramatically improved by the introduction of the novel quantum cascade lasers (QCLs)[4,5]. Yet, diffraction limits the lateral resolution of IR microscopy to the µm range and is, thus, less applicable to examining subcellular components. Spectroscopic techniques that utilize Raman scattering, such as coherent anti-Stokes or stimulated Raman scattering can provide subcellular images at a spatial resolution down to 130 nm[6]. However, due to the high-intensity light source required, Raman spectroscopy is often damaging to biological samples. In contrast, scattering-type scanning near-field optical microscopy (sSNOM) integrates the resolving power of atomic force microscopy (AFM) with the molecular specificity and non-destructive nature of IR spectroscopy. The lateral resolution of sSNOM is not wavelength dependent[7], but dictated by the size of the microscopy probe tip[8], and thus can yield IR images at a resolution down to 5 nm[9]. Employing a broadband IR laser in an interferometric scheme produces nanometer-resolution IR absorption spectra via Fourier-transformed IR spectroscopy (nanoFTIR)[10]. Thus, sSNOM and nanoFTIR present particularly powerful tools for performing molecularly specific measurements of biologically relevant systems, such as single protein complexes[11], individual amyloid fibrils[12] and lipids[13]. In addition to measurements on thin and well defined surfaces, sSNOM and nanoFTIR have been used for the imaging and spectroscopy of whole cells[14]. Recently, a comprehensive database with sSNOM and AFM images of various bacterial species showed the versatility and applicability of near-field imaging in life sciences[15]. The information gathered via traditional sSNOM and nanoFTIR is however limited to around 100 nm below the sample surface[16] and as recently shown, around 200 nm for materials with highly distinguishable contrast from the surrounding[17]. Imaging[18] and chemical identification[19] of subsurface structures and layers have been reported within the penetration depth. Therefore, when examining whole cells with thicknesses in the order of few hundred nanometers to micrometers, internal structures obscured by the cell wall or membrane remain challenging to resolve. Therefore, when examining whole cells with thickness in the order of few hundred nanometers to micrometers, internal structures obscured by the cell wall or membrane remain challenging to resolve.

Our present work describes a nanoscopic technique that applies the super-resolution imaging power and spectroscopic strengths of sSNOM and nanoFTIR to cellular cross sections, prepared by a method well-established in EM. Combining the strengths of these techniques allows for nanoscale resolution, intracellular near-field IR imaging and spectroscopy, to be employed on 100 nm thick

sections of both a prokaryotic and eukaryotic model species. Other recent approaches have applied AFM and nanoFTIR on cross-sectioned plant cells[20]. However, our study expands on the use of mid-IR illumination to report the local protein distribution of internal cellular organelles. Additionally, multiple sequential cross-sections of the same cell were examined, allowing for 3D reconstruction of cellular spectroscopic tomography. Volumetric information from images recorded at various demodulation orders, retrieved via an analytical inversion procedure, was presented in[21] as an initial step towards near-field IR tomography. We present an approach to perform sSNOM tomography by 3D reconstruction of multiple sequential cross-sections with a sum thickness of about ten times the penetration depth. The volume corresponding to the reconstructed tomogram is thus not constrained by the intrinsic limitation of sensitivity in z-direction and can be expanded by increasing the number of sequential cross-sections. This approach enables the detection and visualization of the chemical composition of individual microorganisms and their subcellular components in a non-invasive manner, without the need to include highly specific, potentially artifact-inducing or hazardous sample staining, which can be expanded to imaging of multicellular structures and tissues. Ultimately, IR nanoscopy is able to resolve single protein complexes of a cell as demonstrated in the present work.

## Results

**NanoFTIR and sSNOM operating principle.** Near-field nano-scopy was performed by focusing IR radiation on a metallic AFM tip, generating a confined near field at the tip's apex[22] (Fig. 1a). When the sample is brought close to the tip, light is scattered from the tip and carries information on the optical properties of the surface beneath the tip, where the scattered amplitude relates to the reflectivity and the phase to the absorption of the surface[23]. Essentially, the AFM tip acts as point source of light. As a consequence, optical resolution is defined by the diameter of the tip apex but not by the wavelength. The weak intensity of the scattered photons and the low performance of IR detectors as compared to highly sensitive UV/Vis detectors, is effectively compensated by lock-in detection where the oscillating AFM cantilever provides the carrier frequency for detection. In sSNOM imaging mode, the surface is raster scanned at a single frequency of the QCL thus simultaneously obtaining AFM topography, near-field amplitude and phase images (Fig. 1b). To effectively separate the scattered amplitude and phase information pseudoheterodyne detection scheme[24] is used where the phase is modulated via a piezo-driven mirror in the reference arm of the interferometer. For nanoFTIR spectroscopy, a difference frequency generation-based fs laser[25] is used to illuminate the tip with broadband IR radiation. The scattered light is fed into a Michelson interferometer and a near-field IR absorption spectrum is recorded at each tip position (Fig. 1c). Near-field absorption is then calculated according to $A = \frac{s_n}{s_{n,ref}} \sin(\varphi_n - \varphi_{n,ref})$, where $s_n$ and $s_{n,ref}$ denote the amplitude of the scattered light from the sample and reference, respectively. $\varphi_n$ and $\varphi_{n,ref}$ denote the phase of the scattered light from the sample and reference, demodulated at the n$^{th}$ harmonic of the tip's resonance frequency, thus extracting the near-field contribution of the detected signal[26].

**Subcellular nanoFTIR spectroscopy.** The morphology of *Chlamydomonas reinhardtii* (Fig. 2) and *Escherichia coli* (Supplementary Fig. 1) thin sections were investigated using TEM as a reference to compare against AFM images. Several distinct subcellular features and organelles were identifiable[27] with both TEM (Fig. 2a) and AFM (Fig. 2b), including the nucleus and nucleolus, the photosystem-containing thylakoid membranes, the pyrenoid

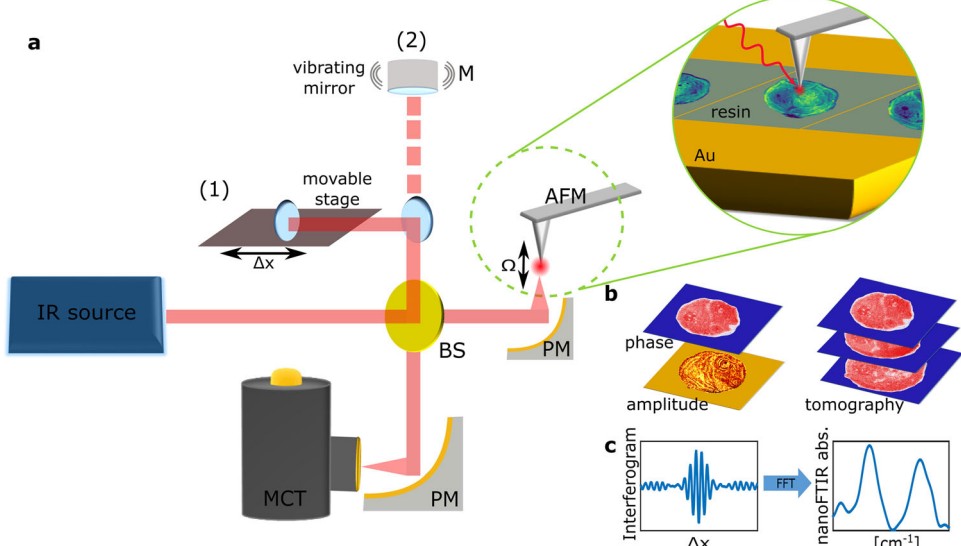

**Fig. 1 Operating modes of IR nanoscopy in an asymmetric Michelson interferometer scheme. a** IR radiation, either broadband fs laser for nanoFTIR or QCL for sSNOM, is guided through a beamsplitter (BS) and focused on an oscillating AFM tip via a parabolic mirror (PM). For nanoFTIR (1), the reference arm mirror is mounted on a movable stage. In sSNOM mode (2), the reference arm is equipped with piezo-driven mirror vibrating with frequency M. The scattered light from the tip is recombined with the reflected light from the reference arm at the BS, focused on a mercury-cadmium-telluride (MCT) detector, and fed to a lock-in amplifier. **b** Simultaneously to AFM imaging, the demodulated sSNOM scattering amplitude and phase are recorded. sSNOM tomography is performed by imaging of serial sections. **c** The detected interferogram is demodulated at sidebands $n\Omega \pm mM$ for harmonics n and m of the tip's resonance frequency $\Omega$ and the mirror's vibration frequency M, respectively. Fast Fourier transformation (FFT) of the interferogram is used to obtain nanoFTIR spectra.

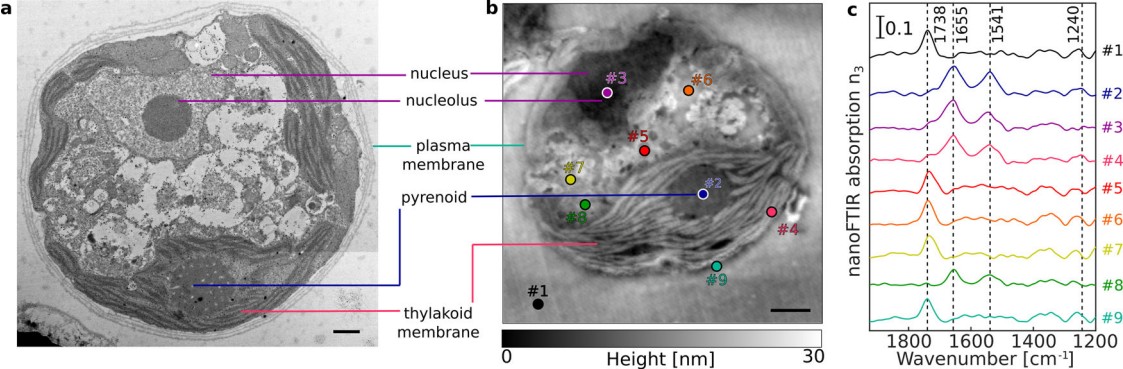

**Fig. 2 Cross-sectional imaging of *C. reinhardtii* cells.** Cellular cross sections were imaged by TEM (**a**) and AFM (**b**). Scale bars equal 1 μm. nanoFTIR absorption spectra (**c**) were acquired at nine different locations marked on the AFM images.

with its distinctive starch inclusions, and the exterior plasma membrane.

Using nanoFTIR, locally resolved spectra were recorded on several intracellular locations as well as on the cell-free region containing only the embedding resin (Fig. 2c, Supplementary Fig. 2b). Each spectrum was measured on a single spot with size given by the AFM tip radius at lateral resolution of 20 nm (Supplementary Fig. 3d), marked on the AFM micrograph and referenced against a spectrum recorded on bare Au surface. The spectral region between 1800 and 1500 cm$^{-1}$ demonstrates multiple characteristic absorption features that vary with different measuring positions (Fig. 2c). Spectrum #1 (black) was measured on a cell-free region and the dominant peak at 1738 cm$^{-1}$ corresponds to the C=O stretching vibration of succinic anhydride present in the epon-like embedding resin. Spectrum #2 (blue) was recorded on the central part of the pyrenoid. It shows distinct absorption peaks at 1655 cm$^{-1}$, assigned to the C=O stretch of amide, and 1540 cm$^{-1}$ stemming from combined

N-H bending and C-N stretching modes, known as amide I and amide II, respectively[11]. These peaks indicate high protein content and, as previously reported, the pyrenoid is largely composed of polypeptides such as RuBisCO[28,29], the key enzyme involved in $CO_2$ fixation. Spectrum #3 (purple) recorded in the nuclear region is dominated by similar absorption of bands at 1655 cm$^{-1}$ and 1540 cm$^{-1}$ suggesting protein content in this region. Yet, the IR bands of DNA (characterized by ring modes of purine and pyrimidine[30,31]) are likely to contribute to this absorption spectrum as well. Spectrum #4 (pink) corresponds to the local IR absorption of the thylakoid with prominent amide I and amide II absorption bands due to the high protein content of the large photosynthetic complexes Photosystem I (PSI) and Photosystem II (PSII) contained within. Additional contributions in the 1650 cm$^{-1}$ and 1520 cm$^{-1}$ regions may be due to the C=O and C=N vibrations of chlorin rings, typical of chlorophyll[32] in PSI and PSII. The low intensity peak at around 1740 cm$^{-1}$ could be related to either chlorophyll or the resin. The cytoplasmic

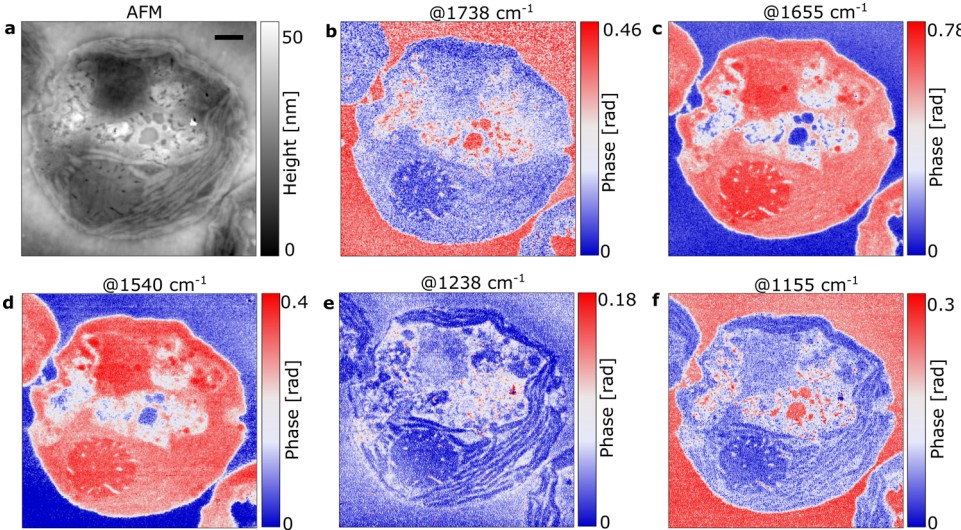

**Fig. 3 sSNOM phase imaging of a *C. reinhardtii* cell.** A single cellular cross section was visualized by AFM topography (**a**) and sSNOM phase imaging at several wavenumbers (**b–f**), with red indicating high and blue indicating low absorption of each respective wavenumber. Scale bar equals 1 μm.

inner region was probed at several locations, each producing similar spectra as observed in the resin (Fig. 2c, spectra #5, #6 and #7), thus indicating resin penetration inside of the cell. In contrast, spectrum #8 (green) has almost no resin absorption but shows distinct amide I and amide II peaks. This suggests that although measured on a visually similar cellular location to spectra #5–7, this spectrum was recorded from a protein-rich organelle. Finally, spectrum #9 (cyan) was taken from the very edge of the cell in order to observe the plasma membrane. However, as the dominant absorption peak at 1738 cm⁻¹ may originate from C=O stretching vibrations of either the membrane lipids or the resin, the assignment remains ambiguous. To gain insight into the near-field absorption at lower wavenumbers, additional measurements were performed covering spectral region down to ~1000 cm⁻¹ (Supplementary Fig. 2) to probe molecular vibrations of phosphate and ribose groups of DNA originating from the nucleus.

**Single-wavelength sSNOM imaging.** Based on our nanoFTIR analysis, we recorded sSNOM images at specific marker wavenumbers. The AFM topography of the selected cell (Fig. 3a) is visually similar to the previously examined sections (Fig. 2a, b), with several distinct morphological features present. Phase-contrast imaging at 1738 cm⁻¹ (Fig. 3b) shows strong absorption in the area surrounding the cell and in several, well-defined intracellular vacuoles, reinforcing the previous identification of the 1738 cm⁻¹ absorption band being primarily due to the embedding resin.

In contrast, Fig. 3c, d show a near complete visual inversion with a uniform lack of absorption across all areas that the resin is present. Similar to nanoFTIR observations (Fig. 2c), visualization at the wavenumbers corresponding to amide I (1655 cm⁻¹, Fig. 3c) and amide II (1540 cm⁻¹, Fig. 3d) both show strong absorption across several cellular features, including the pyrenoid, the chloroplast and the nuclear region. Interestingly, absorption at 1540 cm⁻¹ is relatively homogenous across the nucleus (Fig. 3d), whereas absorption at 1655 cm⁻¹ provides a higher-contrast image, allowing for the nucleolus to be clearly distinguished from the surrounding nuclear area (Fig. 3c). This increase in intensity is likely the result of the high density of proteins, nucleobases and other genetic material in this region.

Imaging at 1238 cm⁻¹ showed low overall absorption (Fig. 3e) with the exception of several small but clearly resolved regions

within the cytoplasm. However, imaging at this wavenumber allowed for clear visualization of the thylakoid stacks within the chloroplast. In addition, the nucleolus is distinguishable from the nucleus with a slight increase in absorption likely due to the asymmetric P-O stretching vibration of phosphate. Cellular imaging at 1155 cm⁻¹ (Fig. 3f) resembles the absorption pattern seen at 1738 cm⁻¹ (Fig. 3b), showing strong absorption on the resin area outside the cell and within cellular vacuoles. Visualization at 1155 cm⁻¹ provides additional contrast, particularly of the thylakoid membranes and the protrusions in and around the pyrenoid. This adds further evidence to the assignment of this absorption feature as C-O-C stretching vibrations, present in a mixture of complex carbohydrates (such as starch) and the embedding resin.

In addition to AFM morphology and sSNOM phase contrast, information on the local reflectivity was gathered via sSNOM scattering amplitude. Several intracellular structures are depicted (Fig. 4), demonstrating both the resolving power of sSNOM imaging and the potential of these imaging modes to each reveal different cellular features. While the cell wall was indistinct when imaged with AFM (Fig. 4a), it is readily apparent when visualized with either amide I phase contrast (Fig. 4b) or scattering amplitude (Fig. 4c), appearing as a distinct ~80 nm thick band with moderate 1655 cm⁻¹ absorption and low reflectance, respectively. The stacking of the thylakoids is most apparent in the scattering amplitude image (Fig. 4c). In contrast, phase imaging provides exceptional detail in the nuclear region allowing for clear differentiation of the nucleolus as well as a spherical nuclear body. A line profile across the nuclear body demonstrated both its particularly high absorption and its diameter of ~200 nm (Fig. 4b insert). As such this nuclear body is likely a Cajal body[33]. The IR response of the cellular components at various wavenumbers is summarized in Supplementary Table 1.

The characteristic architecture of a eukaryotic flagellum can be seen in the axoneme. The scattering amplitude (Fig. 4f) resolves the nine pairs of outer doublet microtubules and their radial spokes, while the central microtubules and the inner sheath can be seen as a slightly lower region of AFM topography (Fig. 4d) in the center of the flagellum. The phase image in contrast, exhibits strong and homogeneous peptide absorption across the flagellum, and moderate absorption of the surrounding cell wall (Fig. 4e). These observed flagella structures are highly comparable to previously reported *Chlamydomonas* flagellum characteristics[34].

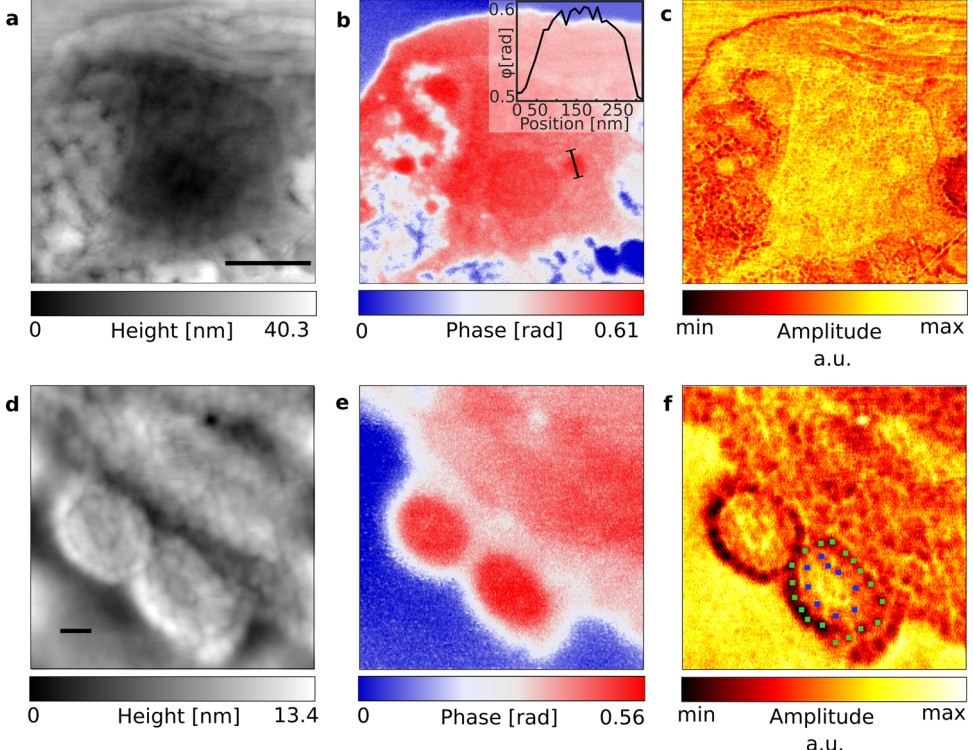

**Fig. 4 Imaging of specific intracellular structures of a *C. reinhardtii* cell.** The nuclear region is visualized by AFM topography (**a**) with the scale bar showing 1 μm, sSNOM phase imaging at 1655 cm⁻¹ (**b**) with the inset showing a line profile across a nuclear body, and sSNOM amplitude (**c**). The axoneme of the flagellum is similarly visualized using AFM topography (**d**) with the scale bar showing 100 nm, sSNOM phase imaging at 1656 cm⁻¹ (**e**) and sSNOM amplitude (**f**), with the doublet microtubules (green marks) and the radial spokes (blue marks) of the flagellum.

The combination of AFM topography and sSNOM spectroscopy was also a valuable tool in the analysis of prokaryotic cells. *E. coli* cross sections (sectioned along either axis) were visually similar when comparing TEM and AFM imaging (Supplementary Fig. 1). Strong amide I and amide II bands are present throughout the cell with strong intracellular resin penetration (Supplementary Fig. 1c and d). Although an *E. coli* cell is approximately ten times smaller than a *C. reinhardtii* cell and does not contain internal membrane-bound vesicles, as evident from the AFM image (Supplementary Fig. 3a), the resolving power of sSNOM imaging at 1658 cm⁻¹ clearly distinguishes several subcellular features. Overall, the cytoplasmic region shows strong absorption at this wavenumber (Supplementary Fig. 3b), but with an anisotropic distribution. A particularly dense region (suggesting high protein content) can be observed in the upper right of the cell. Conversely, there are several central patches of moderate to low amide I absorption. The sSNOM amplitude image is also anisotropic with similar regions varying in intensity (Fig. S2c). The outer membrane has moderate amide I absorption and was used to determine a lateral resolution of ~20 nm of our sSNOM image recordings (Supplementary Fig. 3d).

**sSNOM tomography.** Utilizing the top-view optics of the sSNOM imaging system, it was possible to identify several consecutive cross sections each from the same individual cell. Combining the sSNOM phase images of these ten consecutive, 100 nm thick cellular cross sections (Fig. 5a) resulted in a 1 μm thick 3D tomographic reconstruction (Fig. 5b and Supplementary Movie 1), representing approximately 10% of the entire cell. This tomograph provided a volumetric representation of the local intracellular protein distribution allowing for the 3D visualization of IR signatures of the membrane-bound organelles within a cell. The protein concentration within the pyrenoid is particularly

high and as a result, the contrast in this region demonstrates the arrangement of the starch "fingers" within the pyrenoid. Similarly, the nuclear region also shows strong absorption, however, this dispersal in 3D space is more sporadic than in the pyrenoid. Conversely, the white and blue regions of lower absorption volumetrically represent the size, shape and distribution of the various cellular vacuoles across the cytoplasmic region of the cell. Thus, we infer that sSNOM tomography provides a direct chemical volumetric map at nm resolution where the contrast results from IR absorption.

## Discussion

Near-field nanoscopy was employed as a method to chemically image biological samples, specifically the internal cellular structures of both prokaryotic (*E. coli*) and eukaryotic (*C. reinhardtii*) model species. NanoFTIR spectroscopy provided data from the mid-IR region, and sSNOM imaging was utilized to map this chemical data with a resolution well below the diffraction limit of about 5 μm. The level of chemical detail achieved with sSNOM phase imaging goes beyond what is available through TEM, and is further supplemented by the simultaneously gathered AFM and sSNOM amplitude micrographs. Additionally, sSNOM phase imaging of several consecutive cross sections of a single cell allowed us to produce a 3D volumetric image of local intracellular protein distribution with dimensions of each voxel being 20 nm × 20 nm × 1 μm corresponding to 400 zeptoliter ($4 \times 10^{-19}$ l). The preparation of cellular samples by resin embedding and cross sectioning shown here, can be readily applied in photothermal-expansion microscopy, such as AFM-IR[35].

Nanoscale IR imaging and tomography is an attractive new approach to subcellular label-free imaging, which complements the growing arsenal of other super-resolution imaging modalities. One such complementary vibrational technique to IR nanoscopy

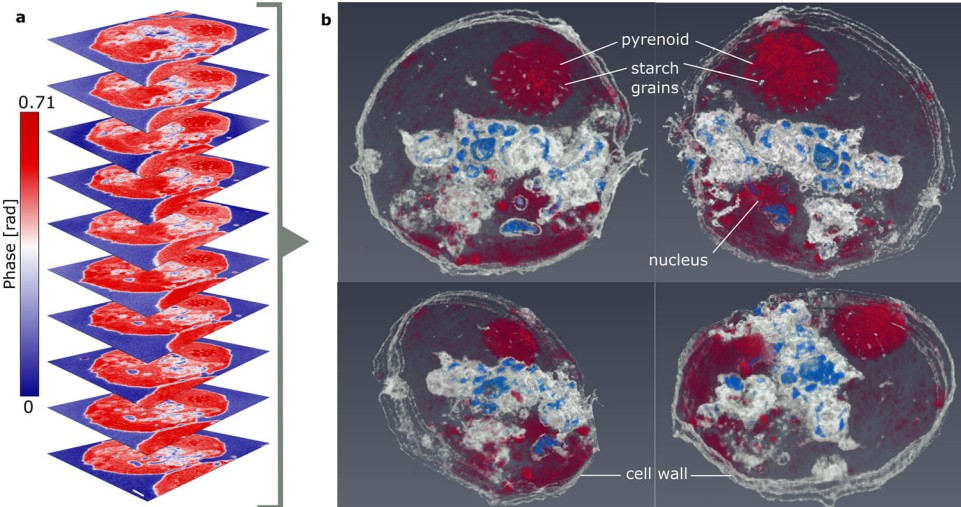

**Fig. 5 sSNOM tomography of a *C. reinhardtii* section.** Ten sSNOM images of consecutive *C. reinhardtii* cross sections, recorded at 1656 cm$^{-1}$ (**a**) used for the construction of a tomogram (**b**), shown in four orientations: top right—view from the top, top left — view from the bottom, bottom left—top view tilted, bottom right - bottom view tilted (Supplementary Movie 1). Scale bar equals 1 µm.

is tip-enhanced Raman scattering (TERS). This technique can resolve single molecules[36]. Detection of vibrational spectra of biological samples (such as proteins and genetic material) is hampered by the weak Raman scattering cross section and the required high laser intensities may lead to sample damage. High-resolution TERS imaging of biological samples such as whole cells or cellular cross-sections at cryogenic temperatures and by using super-sharp tips, is an ongoing challenge[37].

Fluorescence microscopy offers excellent resolving power and can be employed to identify a multitude of cellular features, physiological properties and various metabolic activities, especially of living cells[38]. Fluorescence microscopy offers superior imaging contrast, yet fluorescence spectra are broad and the application of a number of dyes for spectral multiplexing is limited. Here, the vibrational spectra of biological samples provide a manifold of characteristic bands for multiplexing. Hyperspectral IR nanoscopy[39] is still in its infancy but technical progress in concert with the application of machine-learning and deep-learning algorithms is expected to provide unprecedented chemical contrast[40]. Unlike fluorescence-based super-resolution imaging techniques, sSNOM presents an approach to nanoscale chemical imaging that does not rely on the use of chemical labeling, and can examine prepared samples at room temperature in a non-destructive manner. While fluorescence emission degrades over time, IR spectroscopy is minimal invasive and the presented sample preparation method provides specimens that are stable over extended periods.

The spatial resolution in sSNOM is proportional to the AFM tip size. Thus, decreasing the apex radius directly improves the achievable resolution but on the expense of the reflected signal intensity. While sSNOM shows high sensitivity at high spatial resolution due to nonlinear dependence of the detected intensity on the near-field enhancement, the power dependence of the incident light is the same as in conventional far-field microscopy[41]. Conceptually, this relates to the modified Abbe's equation describing the diffraction limit where the intensity of the focal area in stimulated emission depletion (STED) microscopy is taken into account[42].

Through the combination of near-field spectroscopy and imaging, several specific inferences can be made about the examined samples. Phase-contrast imaging proved to be specifically useful in identifying areas of condensed protein or genetic material. For example, IR nanoscopy revealed regions likely to be rich in genetic material in both *E. coli* (Supplementary Fig. 2c)

and *C. reinhardtii* (Figs. 3c and 4b). The Cajal body identified in *C. reinhardtii* (Fig. 4b) is typically found in proliferating cells. Therefore, there is potential that sSNOM could be used to identify physiologically or metabolically active cells.

Expanding sSNOM imaging to a spectral range down to 800 cm-1 is valuable for probing the nuclear area via the asymmetric phosphate stretching vibration, deoxyribose C-O stretching and O-P-O bending. This, however, remains challenging due to the low emission of the IR source used in this spectral region. Utilizing IR radiation at wavenumbers higher than 2500Expanding sSNOM imagingcm-1 could add another level of contrast detail by probing the amide A and B vibrations of the peptide bond[43], the C-H stretching vibrations of membrane lipids[44] and of the O-H stretching vibrations of (bound) water molecules[45].

NanoFTIR spectroscopy of the inner cytoplasmic region of *C. reinhardtii* showed several areas of resin penetration, of which the spatial orientation was identified through sSNOM imaging. Resin in these areas suggests that water-soluble cellular components were replaced by resin during the embedding process. Furthermore, the distinct layered structure of the thylakoids was clearly resolved, but the chlorophyll peaks typical of chloroplasts, were not clearly distinguished in the nanoFTIR spectra due to rather low absorption intensity. This is not unexpected however, as samples require dehydration with alcohol prior to resin embedding, visibly leaching green pigments from the microalgae samples.

As such, the resin-embedding process may introduce artifacts. However, current advances in IR nanoscopy can be integrated, further enhancing outputs and expanding operability into more biologically relevant conditions and more extreme environments. For instance, sSNOM imaging has been performed on cryo-sectioned biological tissue samples[46], thus avoiding the need for resin embedding and the commonly associated artifacts. Non-embedded samples can also be subsequently stained, allowing for correlative fluorescent imaging. Additionally, imaging sensitivity could be increased even further by performing sSNOM imaging directly under cryogenic conditions[47]. Furthermore, sSNOM imaging has recently been performed on thin protein-rich membranes[48] and peptoid nanosheets[49] immersed in water. Whilst resin embedding is a well-established technique that innately prevents sample degradation (and thus enabled sSNOM tomographic analysis of consecutive cellular cross sections), integration of alternate sample preparation techniques has the

potential to immensely further the bio-imaging applications of IR nanoscopy.

The IR tomographical technique described here can be expanded upon and potentially used alongside automation techniques typical for TEM tomography. The 3D chemical mapping of an entire cell and its substructures is a logical advance which could be adapted to studying even multicellular structures and tissue. We also envisage this method to be further applied to subcellular imaging, specifically investigating the function and structural details of individual cellular components through 3D visualization of IR absorption.

The potential spectroscopic information that can be gathered with this technique may be remarkably complex with a multitude of organic chemical groups contributing to the absorption in the IR region. While the focus of the presented work lies on the analysis of raw sSNOM phase and amplitude, additional information can be retrieved from the material's optical constants[15,50]. We may infer that applying machine learning to multichannel datasets of IR and topographical images will be crucial to extract more detailed information from cellular cross-sections[51]. Therefore, insights gathered via single-wavelength sSNOM imaging will be extremely valuable, furthering the possibility of deeply exploring the molecular composition of individual microorganisms and their subcellular components. In essence, super-resolution IR imaging and tomography provides a broad potential for major contributions to the field of subcellular biology including nanomedicine, nanotoxicology, disease-related imaging, diagnostics and therapy prediction.

## Methods

**Cell cultures preparation and resin embedding**. The green alga *C. reinhardtii* CC-125 (mt+) was cultured under constant illumination of 50 μmol photons m$^{-2}$ s$^{-1}$, agitation of 110 rpm at 28 °C in Tris-Acetate-Phosphate medium[52]. *E. coli* K12 (DSM 498) was cultured with 110 rpm of agitation at 37 °C in Brain Heart Infusion Broth (Merck, Germany). After reaching log-phase growth, cultures were fixed in Cacodylate combination fixative (0.1 M Cacodylate buffer, 2% formaldehyde, 2.5% glutaraldehyde, 0.088 M sucrose, 0.001 M CaCl$_2$.H$_2$O, pH adjusted to 7.4; Merck, Germany) at room temperature for two hours. Following fixation, aliquots of each culture for TEM imaging were stained for 1 h in 1% osmium tetroxide (diluted in 0.1 M Cacodylate buffer). All samples were dehydrated using an ethanol series from 50–100% (v/v), increasing in 10% increments and incubating at room temperature for 30 min at each concentration. Following the 100% ethanol incubation, each sample was resuspended twice in propylenoxid (Merck, Germany).

Samples were centrifuged for 3 min, 14,000 rcf, and the resulting pellet was maintained as well as possible during the resin-embedding process. Agar Low Viscosity Resin (Agar Scientific, UK) was prepared as per manufacturer's instructions, and then added to each sample at a 50% concentration (v/v in ethanol). Samples were incubated at room temperature overnight, in open vessels, to encourage evaporation. The resin concentration was then increased in 10% increments, incubating at room temperature for 30 min with each resin addition. Following an additional resuspension in 100% resin, samples were placed under vacuum for 30 min, before curing at 60 °C overnight.

**Thin-sections preparation and TEM imaging**. Resin-embedded samples were cut to a thickness of 100 nm, using a microtome (Reichert ultracut S, Leica, Germany) fitted with a diamond knife (Histo 45°, DiATOME, USA). The ultrathin sections for TEM imaging were mounted on nickel square mesh grids (AGG2200N, Agar scientific, UK). These samples were subsequently stained with UranyLess (Delta Microscopies, France) for 2 min, followed by 3% Reynolds lead citrate (Delta Microscopies, France) for 2 min. To prevent precipitation of the lead citrate, pellets of NaOH were added to the staining chamber to exclude atmospheric CO$_2$. Samples were dried and washed at least three times with double distilled water after each step. TEM was performed using a JEM-1400Flash (JEOL, Japan). For image acquisition and processing, software provided by the TEM manufacturer was used (TEM Center 1.7.18.2349, 2006-20018 JEOL, Japan). The microscope was used with high voltage of 120 kV with a beam current of 60.80 μA, while using a medium spot-size to reduce the risk of damaging the section. A 9 × 9 image montage was recorded at high magnification, and after brightness correction, was compiled into one stitched image.

**AFM, nanoFTIR and sSNOM sample preparation and experimentation**. Thin sections for AFM, nanoFTIR and sSNOM analysis were transferred to a flat template stripped gold (TS Au) surface. To prepare the TS Au surfaces[53], a 100 nm thick gold (99.999% Au, Kurt J. Lesker Company, Germany) layer was thermally evaporated onto a silicon wafer using physical vapor deposition (MBraun, Germany). Microscopy glass slides cut in ~1 × 1 cm pieces were glued on the Au surface using adhesive (NOA 81, Norland Products, USA) and cured with UV lamp. The glass pieces were then removed from the silicon surface and inverted to obtain a flat gold surface for sample deposition.

In order to gather consecutive cross sections for sSNOM tomographic analysis, prior to ultrathin sectioning, the resin-embedded samples were cut into trapezoidal shape, thus causing sequential thin sections to adhere to the previous section, thus maintaining their orientation by forming a ribbon. This ribbon was then transferred to a TS Au surface as above.

An Atomic Force Microscope (NanoWizard II, JPK Instruments, Bruker Nano, Germany) was used for imaging of the thin sections under ambient conditions. The microscope was operated in tapping mode, using commercially available tips (Arrow NCPt, NanoWorld, Switzerland) with resonance frequency of 285 kHz. NanoFTIR spectroscopy was performed using a custom-built setup[14], equipped with broadband femtosecond laser source (FemtoFiber dichro midIR, NeaSpec, Germany). The sample side of the recorded interferograms was Fourier transformed to obtain the amplitude and phase spectra. Each spectrum represents an average of 50 co-additions measured at spectral resolution of 16 cm$^{-1}$. For sSNOM imaging, a commercial system (neaSNOM, NeaSpec, Germany) with four quantum cascade laser chips (Daylight Solutions, USA) covering the mid-IR spectral region was used. The AFM and sSNOM images were visualized using Gwyddion (v. 2.51). 3D reconstruction and visualization was performed using Amira 6.3.0 (Avizo, ThermoFisher Scientific, Germany).

**Reporting summary**. Further information on research design is available in the Nature Research Reporting Summary linked to this article.

## Data availability

The datasets generated during and/or analyzed during the current study are deposited in the repository Figshare. https://doi.org/10.6084/m9.figshare.16750555, https://doi.org/10.6084/m9.figshare.16782391, https://doi.org/10.6084/m9.figshare.16782403, https://doi.org/10.6084/m9.figshare.16782406, https://doi.org/10.6084/m9.figshare.16802113. Microscopic images are deposited in BioIMage Achive with accession code S-BIAD194.

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

## Acknowledgements

This work has been funded by a grant from the German Research Foundation to J.H. (DFG, project HE 2063/5-1) and to D.J.N (Emmy-Noether program, NU 421/1-1). A.E. was supported by the European Union's Horizon 2020 research and innovation program under the Marie Skłodowska-Curie grant agreement No. 706072 (NanoMembR). D.J.B. and A.E. gratefully acknowledge funding by BMWi/DLR (grant number 50WB1623 and 50WB2023). A.E. acknowledges funding by Volkswagen Foundation and its Freigeist Program. The authors acknowledge support by the research building SupraFAB of Freie Universität Berlin.

## Author contributions

The project idea was conceived and coordinated by A.E. and J.H. Microalgae cultures were provided by D.J.N. while sample preparation and resin embedding were performed by D.J.B. Microtomy and mounting of serial sections was performed by A.E. and D.J.B. AFM, nanoFTIR and sSNOM experiments were performed by K.K. Spectroscopic analysis was supported by J.H. 3D reconstruction was performed by K.K. and image interpretation was supported by D.J.B. and D.J.N. Counterstaining and TEM imaging was contributed by P.K.H. The manuscript was written by K.K., D.J.B. and J.H. with contributions by the other authors. All authors read and acknowledged the manuscript.

## Funding

## Competing interests

The authors declare no competing interests.
