## [Peer Review File · Communications Biology]

Reviewers' comments:

Reviewer #1 (Remarks to the Author):

The authors introduce a method for volumetric imaging of biological samples that relies on s-SNOM imaging and nanoFTIR spectroscopy of thin sections prepared according to a TEM protocol. They show that s-SNOM images collected for the sequentially cut sample sections can be processed to result in a 3D representation of the sample, similar to those that can routinely be obtained in the case of imaging techniques with optical sectioning capabilities. I find the manuscript to be well written, and the work described in sufficient level of detail to be reproduced. The impact of the work is important, but, while s-SNOM is discussed as a potential replacement for fluorescence based techniques (the authors referring here to aspects such as resolution or use of endogenous contrast), this can be possible only in very specific applications (for example I imagine that s-SNOM imaging of live cells is difficult to say the least, due to the acquisition speed; furthermore, imaging cells without sectioning them is hindered by the shallow penetration depth of s-SNOM). In my opinion, s-SNOM can rather be seen as a very important tool to complement fluorescence techniques, by making available additional information not available to these, thus I consider the proposed work to be important. Below, I list a series of suggestions/concerns:

1) The opening line of the Abstract states:

"The few microscopic techniques that simultaneously gather morphological and chemical data often rely on the use of specific markers."

Which are these "few microscopic techniques that simultaneously gather morphological and chemical data" based on specific markers? I invite the authors to nominate these or rephrase.

2) In the introduction the authors state:

"The information gathered via traditional sSNOM and nanoFTIR is however limited to a few tens of nm below the sample surface [15]".

Higher imaging depth has been reported though, see for example:

Wenhao Zhang and Yuhang Chen, "Visibility of subsurface nanostructures in scattering-type scanning near-field optical microscopy imaging," *Opt. Express* 28, 6696-6707 (2020).

Other studies reporting sub-surface s-SNOM imaging (e.g., Mester L, Govyadinov AA, Chen S, Goikoetxea M, Hillenbrand R. Subsurface chemical nanoidentification by nano-FTIR spectroscopy. *Nature communications*. 2020 Jul 3;11(1):1-0; Jung, L., Hauer, B., Li, P., Bornhöfft, M., Mayer, J. and Taubner, T., 2016. Exploring the detection limits of infrared near-field microscopy regarding small buried structures and pushing them by exploiting superlens-related effects. *Optics express*, 24(5), pp.4431-4441.; etc.), which is connected to the penetration depth of s-SNOM, have been as well reported. I recommend to the authors to write a brief paragraph summarizing these past efforts. Works on s-SNOM nanotomography should be discussed as well to provide a larger picture of s-SNOM capabilities for volumetric imaging (e.g. Govyadinov, Alexander A., et al. "Recovery of permittivity and depth from near-field data as a step toward infrared nanotomography." *Acs Nano* 8.7 (2014): 6911-6921; Wang, Haomin, et al. "Tomographic and multimodal scattering-type scanning near-field optical microscopy with peak force tapping mode." *Nature communications* 9.1 (2018): 1-11.

3) The description of s-SNOM's working principles on lines 79-82 is too simplistic. The authors should elaborate, so that potential readership from the biology field can get a good better understanding of the underlying principles of this technique, without having to navigate to other publications.

4) Fig 1, the frequency of the vibrating mirror is also involved in the lock-in detection scheme. The authors should revise/amend the caption accordingly.

5) Lines 257-260: "Unlike fluorescence-based super-resolution imaging techniques, sSNOM presents an approach to nanoscale chemical imaging that does not rely on the use of chemical labeling, and can be applied to samples under ambient conditions. "

This sentence is misleading, I recall that the authors had to cut sections of the sample to be able to image features that wouldn't have been accessible in an intact sample due to the s-SNOM penetration depth. Thus mentioning "ambient conditions" is confusing. The authors should revise or elaborate on which are the specific scenarios where s-SNOM can bring added value over fluorescence based techniques.

6) Lines 283-287: "As such, the resin-embedding process may be introducing artifacts into 283 the samples. However, current 284 advances in IR nanoscopy could be integrated, further enhancing outputs and expanding operability into 285 more biologically relevant conditions and more extreme environments. For instance, cryo-sectioning of 286 samples would circumvent the need for resin embedding and performing sSNOM imaging under 287 cryogenic conditions could increase sensitivity even further [38]."

s-SNOM has already been used before to image (tissue) sections prepared according to a TEM protocol that involves cutting thin sections of a sample frozen in liquid nitrogen (Stanciu, S.G., et al. "Correlative imaging of biological tissues with apertureless scanning near-field optical microscopy and confocal laser scanning microscopy." *Biomedical optics express* 8.12 (2017): 5374-5383). In this past experiment, to avoid resin embedding, which can contribute to resin induced roughness and other artefacts, a thin layer of dried methylcellulose was applied over the sample surface. Such past efforts, and their advantages/disadvantages compared to the proposed sample preparation methodology should be discussed.

7) It would be useful if a table is introduced to summarize what cell components respond (yield contrast) for the wavelengths that have been used in this experiment. Also, it would be interesting if the authors can carry a discussion in the main text on what additional wavelengths would be useful to bring additional information with respect to understanding the samples here addressed (and eventually other bacteria species).

8) The authors should provide a brief literature survey on s-SNOM imaging of bacteria.

9) Fig 1 in the supplementary is important, and the authors should consider promoting it to the main text.

10) Besides raw s-SNOM amplitude or phase images, it has been demonstrated to date that s-SNOM can also yield quantitative information on the refractive index/dielectric function of an investigated sample. This was discussed recently also in the context of bacteria. It would be interesting if the authors would discuss how this additional capacity can augment their method.

Reviewer #2 (Remarks to the Author):

In this manuscript by Kanevche et al., the authors present a new route to achieve nanoscale chemical imaging with tomographic capability on intracellular structures. The idea of combining s-SNOM with tomographic sectioning & reconstruction is novel, and the demonstration is satisfactory. This work opens a route toward a detailed understanding of the inner organization of cellular structures at the nanoscale with 20 nm spatial resolution, without labels. I regard this work as a high-impact, transforming work in the applications of infrared s-SNOM. Therefore, I recommend it to publish without delays.

Please find below our point-by-point response (in plain text) to the reviewers' comments (*italics*). The modifications to the manuscript are highlighted in red.

Reviewer 1:

1) *The opening line of the Abstract states: "The few microscopic techniques that simultaneously gather morphological and chemical data often rely on the use of specific markers."*

Which are these "few microscopic techniques that simultaneously gather morphological and chemical data" based on specific markers? I invite the authors to nominate these or rephrase.

We have accordingly rephrased the first sentence in the Abstract, including two examples of specific markers (localized and chemically-specific).

Although techniques such as fluorescence-based super-resolution imaging or confocal microscopy simultaneously gather both morphological and chemical data, these techniques often rely on the use of localized and chemically specific markers.

2) *In the introduction the authors state: "The information gathered via traditional sSNOM and nanoFTIR is however limited to a few tens of nm below the sample surface [15]". Higher imaging depth has been reported though, see for example: Wenhao Zhang and Yuhang Chen, "Visibility of subsurface nanostructures in scattering-type scanning near-field optical microscopy imaging," Opt. Express 28, 6696-6707 (2020). Other studies reporting sub-surface s-SNOM imaging (e.g., Mester L, Govyadinov AA, Chen S, Goikoetxea M, Hillenbrand R. Subsurface chemical nanoindentification by nano-FTIR spectroscopy. Nature communications. 2020 Jul 3;11(1):1-0; Jung, L., Hauer, B., Li, P., Bornhöfft, M., Mayer, J. and Taubner, T., 2016. Exploring the detection limits of infrared near-field microscopy regarding small buried structures and pushing them by exploiting superlens-related effects. Optics express, 24(5), pp.4431-4441.; etc.), which is connected to the penetration depth of s-SNOM, have been as well reported. I recommend to the authors to write a brief paragraph summarizing these past efforts.*

While subsurface sSNOM imaging and nanoFTIR spectroscopy has been shown, detection of structures below the surface was possible within the penetration depth. In the case of *C. reinhardtii*, algae cell with size in the order of 10 μm , accessing the inner structure is still challenging. However, this comment remains valid and we appreciate the chance to expand the limiting penetration depth argument and refer accordingly to novel studies by adding the following to Introduction:

The information gathered via traditional sSNOM and nanoFTIR is however limited to around 100 nm below the sample surface [16] and as recently shown, around 200 nm for materials with highly distinguishable contrast from the surrounding [17]. Imaging [18] and chemical identification [19] of subsurface structures and layers have been reported within the penetration depth. Therefore, when examining whole cells with thicknesses in the order of few hundred nanometers to micrometers, internal structures obscured by the cell wall or membrane remain challenging to resolve.

Works on s-SNOM nanotomography should be discussed as well to provide a larger picture of s-SNOM capabilities for volumetric imaging (e.g. Govyadinov, Alexander A., et al. "Recovery of permittivity and depth from near-field data as a step toward infrared nanotomography." Acs Nano 8.7 (2014): 6911-6921; Wang, Haomin, et al. "Tomographic and multimodal scattering-type scanning near-field optical microscopy with peak force tapping mode." Nature communications 9.1 (2018): 1-11.

We agree with the reviewer that the first step towards nanoscopic tomography was indeed outlined in the paper by Govyadinov et al., ACS Nano (2014). Although the approach to

reconstruct tomography from the various harmonics differs from our “full” tomographic approach by measuring each layer, we appreciate the reviewer’s suggestion to refer to this very important work. Accordingly, we added the following paragraph in the Introduction:

Volumetric information from images recorded at various demodulation orders, retrieved via an analytical inversion procedure, were presented in [21] as an initial step towards near-field IR tomography. We present an approach to perform sSNOM tomography by 3D reconstruction of multiple sequential cross-sections with a sum thickness of about ten times the penetration depth. The volume corresponding to the reconstructed tomogram is thus not constrained by the intrinsic limitation of sensitivity in z-direction and can be expanded by increasing the number of sequential cross-sections.

3) *The description of s-SNOM's working principles on lines 79-82 is too simplistic. The authors should elaborate, so that potential readership from the biology field can get a good better understanding of the underlying principles of this technique, without having to navigate to other publications.*

We agree and expanded the sSNOM and nanoFTIR description by adding the following paragraphs:

To effectively separate the scattered amplitude and phase information a pseudoheterodyne detection scheme [24] is used where the phase is modulated via a piezo-driven mirror in the reference arm of the interferometer. (...)

Near-field absorption is then calculated according to $A = \frac{s_n}{s_{n,ref}} \sin(\varphi_n - \varphi_{n,ref})$, where s_n and $s_{n,ref}$ denote the amplitude of the scattered light from the sample and reference, respectively. φ_n and $\varphi_{n,ref}$ denote the phase of the scattered light from the sample and reference, demodulated at the n^{th} harmonic of the tip’s resonance frequency, thus extracting the near-field contribution of the detected signal [26].

4) *Fig 1, the frequency of the vibrating mirror is also involved in the lock-in detection scheme. The authors should revise/amend the caption accordingly.*

We agree and modified Figure 1 and its caption accordingly:

In sSNOM mode (2), the reference arm is equipped with piezo-driven mirror vibrating with frequency M . (...) The detected interferogram is demodulated at sidebands $n\Omega \pm mM$ for harmonics n and m of the tip’s resonance frequency Ω and the mirror’s vibration frequency M , respectively.

5) *Lines 257-260: "Unlike fluorescence-based super-resolution imaging techniques, sSNOM presents an approach to nanoscale chemical imaging that does not rely on the use of chemical labeling, and can be applied to samples under ambient conditions."*

This sentence is misleading, I recall that the authors had to cut sections of the sample to be able to image features that wouldn't have been accessible in an intact sample due to the s-SNOM penetration depth. Thus mentioning "ambient conditions" is confusing.

The authors should revise or elaborate on which are the specific scenarios where s-SNOM can bring added value over fluorescence based techniques.

We agree that the use of “ambient” in this context is not appropriate. We have therefore modified the following sentence:

Unlike fluorescence-based super-resolution imaging techniques, sSNOM presents an approach to nanoscale chemical imaging that does not rely on the use of chemical labeling, and can examine prepared samples at room temperature in a non-destructive manner. While fluorescence emission degrades over time, IR spectroscopy is minimal invasive and the presented sample preparation method provides specimens that are stable over extended periods.

6) Lines 283-287: "As such, the resin-embedding process may be introducing artifacts into the samples. However, current advances in IR nanoscopy could be integrated, further enhancing outputs and expanding operability into more biologically relevant conditions and more extreme environments. For instance, cryo-sectioning of samples would circumvent the need for resin embedding and performing sSNOM imaging under cryogenic conditions could increase sensitivity even further [38]."

s-SNOM has already been used before to image (tissue) sections prepared according to a TEM protocol that involves cutting thin sections of a sample frozen in liquid nitrogen (Stanciu, S.G., et al. "Correlative imaging of biological tissues with apertureless scanning near-field optical microscopy and confocal laser scanning microscopy." *Biomedical optics express* 8.12 (2017): 5374-5383). In this past experiment, to avoid resin embedding, which can contribute to resin induced roughness and other artefacts, a thin layer of dried methylcellulose was applied over the sample surface. Such past efforts, and their advantages/disadvantages compared to the proposed sample preparation methodology should be discussed.

We agree that the mentioned reference is highly relevant and as such, this paragraph has been extended upon to include a comparison of the reference with the resin-embedding method:

For instance, sSNOM imaging has been performed on cryo-sectioned biological tissue samples [45], thus avoiding the need for resin embedding and the commonly associated artefacts. Non-embedded samples can also be subsequently stained, allowing for correlative fluorescent imaging. Additionally, imaging sensitivity could be increased even further by performing sSNOM imaging directly under cryogenic conditions [46]. (...) Whilst resin embedding is a well-established technique that innately prevents sample degradation (and thus enabled sSNOM tomographic analysis of consecutive cellular cross sections), integration of alternate sample preparation techniques has the potential to immensely further the bio-imaging applications of IR nanoscopy.

7) It would be useful if a table is introduced to summarize what cell components respond (yield contrast) for the wavelengths that have been used in this experiment.

We appreciate the suggestion that a table summarizing the IR response of the cell components complements the description of our observations. Therefore, we added Table 1 to the supplementary information and referred to it in the main text.

Also, it would be interesting if the authors can carry a discussion in the main text on what additional wavelengths would be useful to bring additional information with respect to understanding the samples here addressed (and eventually other bacteria species).

This is an excellent point, motivating an outlook for experiments in the future. Indeed, several molecular vibrations relevant for the cells investigated here show absorption in the high wavenumber region ($>2500\text{ cm}^{-1}$), as well as between $\sim 1000\text{ cm}^{-1}$ and 800 cm^{-1} . We addressed this suggestion by adding the following paragraph to the discussion:

Expanding sSNOM imaging to a spectral range down to 800 cm^{-1} is valuable for probing the nuclear area via the asymmetric phosphate stretching vibration, deoxyribose C-O stretching and O-P-O bending. This, however, remains challenging due to the low emission of the IR source used in this spectral region. Utilizing IR radiation at wavenumbers higher than 2500 cm^{-1} could add another level of contrast detail by probing the amide A and B vibrations of the peptide bond [43], the C-H stretching vibrations of membrane lipids [44] and of the O-H stretching vibrations of (bound) water molecules.

8) The authors should provide a brief literature survey on s-SNOM imaging of bacteria.

While we do not solely focus on bacterial cells, sSNOM imaging on whole bacterial species is necessarily very closely linked to our work. This was addressed in the introduction as follows:

In addition to measurements on thin and well defined surfaces, sSNOM and nanoFTIR have been used for the imaging and spectroscopy of whole cells [14]. Recently, a comprehensive database with sSNOM and AFM images of various bacterial species showed the versatility and applicability of near-field imaging in life sciences [15].

9) *Fig 1 in the supplementary is important, and the authors should consider promoting it to the main text.*

Indeed, the supplementary figure Figure S1 carries important information on the morphology and near-field spectra of *E. coli* sections. However, the topic of the current work is to resolve subcellular structures which is evidently not present in bacterial cells lacking compartmentalization. Therefore, we decided to present the experiments on *C. reinhardtii* as a showcase for resolving the complex subcellular structure including membrane-bound organelles. For the sake of readability and flow of the main text, we believe that our studies on *E. coli* (both sSNOM and nanoFTIR) fit better to the supplementary information.

10) *Besides raw s-SNOM amplitude or phase images, it has been demonstrated to date that s-SNOM can also yield quantitative information on the refractive index/dielectric function of an investigated sample. This was discussed recently also in the context of bacteria. It would be interesting if the authors would discuss how this additional capacity can augment their method.*

We definitely agree with the reviewer on this important aspect. Prominent contrast was observed in the sSNOM phase of our images, thus enabling us to distinguish regions even within the nuclear area, such as the Cajal body. Looking closely into the sSNOM amplitude, on the other hand, revealed the flagellum inner structure, which otherwise showed rather homogenous phase contrast. This example highlights the potential benefits of considering various channels such as the topography and especially the wavelength-dependent complex optical properties of the sample. Motivated by the comment of the reviewer, we included the following paragraph in the discussion:

While the focus of the presented work lies on the analysis of raw sSNOM phase and amplitude, additional information can be retrieved from the material's optical constants [15],[48]. We may infer that applying machine learning to multichannel datasets of IR and topographical images will be crucial to extract more detailed information from cellular cross-sections.

Reviewer 2:

In this manuscript by Kanevche et al., the authors present a new route to achieve nanoscale chemical imaging with tomographic capability on intracellular structures. The idea of combining s-SNOM with tomographic sectioning & reconstruction is novel, and the demonstration is satisfactory. This work opens a route toward a detailed understanding of the inner organization of cellular structures at the nanoscale with 20 nm spatial resolution, without labels. I regard this work as a high-impact, transforming work in the applications of infrared s-SNOM. Therefore, I recommend it to publish without delays.

We are delighted about the enthusiastic comments of the reviewer.